# To Meet, to Matter, and to Have Fun: The Development, Implementation, and Evaluation of an Intervention to Fulfil the Social Needs of Older People

**DOI:** 10.3390/ijerph16132307

**Published:** 2019-06-28

**Authors:** Tina ten Bruggencate, Katrien G. Luijkx, Janienke Sturm

**Affiliations:** 1Department of Tranzo, School of Social and Behavioral Sciences, Tilburg University, 5037 AB Tilburg, The Netherlands; 2Chair of People and Technology, Institute for HRM and Psychology, Fontys University of Applied Science, 5612 AR Eindhoven, The Netherlands

**Keywords:** social needs, older adults, interventions, volunteer work

## Abstract

Interventions for older people are often not evaluated and, if evaluated, are not proven successful. Based on a systematic literature review and two qualitative studies about the social needs of older people, an intervention has been developed, implemented, and evaluated. Important social needs that emerged from these studies are connectedness, meaningfulness, and independence. Samsam, the developed intervention, aims to fulfil these needs. Samsam is a place where older (native Dutch speaking) people teach the Dutch language to expats, refugees, and immigrants. Two group interviews and one interview with a total of seven older participants were held to find out what the experiences are with this intervention to fulfil the social needs of older people. After analysis, three themes emerged: (1) The general experience of Samsam, (2) connectedness, and (3) meaningfulness and status. Results indicate that the volunteers are content with the conditions of the intervention, although it is sometimes hard work. The older participants indicated that helping other people and contributing to society is important for them. The intervention also has a strong social and fun element which contributes to their feeling of connectedness. The intervention fulfils various social needs, such as connectedness, meaningfulness, and status. When participating in Samsam, participants feel connected to each other, to the students, and to society. The older participants want to have meaningful lives and use their skills and talents. Samsam offers possibilities for them to do so. We further found that participants have some common characteristics such as an openness to others and to other cultures. An openness towards others and to society helps older people to connect. Most interventions focus on stimulating contact between older people, primarily on their need for affection. We conclude that meaningfulness and status are important social needs. Successful interventions for older people should focus more on fulfilling these needs—for example, by engaging in purposeful activities. It becomes easier to connect when a person feels useful.

## 1. Introduction


*“When you are old, you have all the answers, but nobody asks you the questions.”*
*(Dr. Laurence J. Peter)*

### 1.1. Social Needs and Well-Being

Social needs are important needs for every individual [1,2,3]. Rowe and Kahn [4] stated that besides the low probability of disease and high cognitive and physical functional capacities, social engagement is one of three major elements of successful ageing. They define social engagement in two ways: (1) Remaining involved in activities that are meaningful and purposeful, as well as (2) maintaining close relationships. Older people value their social life and social well-being even more than their physical or cognitive functioning [5]. Steverink and Lindenberg [3] identified, in their social production function theory, three social needs: Affection, behavioral confirmation, and status. Affection is fulfilled by relationships that give you the feeling that you are liked, loved, trusted, and accepted. Behavioral confirmation is fulfilled by relationships that give you the feeling of doing the right thing in the eyes of relevant others and yourself. Status is fulfilled by relationships that give you the feeling that you are being treated with respect, are being taken seriously and are independent or autonomous. Status also refers to being known for your achievements, skills, or assets. A study by Bruggencate [6] found that the social needs that are relevant for older people are connectedness, meaningfulness, and independence. These three needs largely correspond with the three needs of the aforementioned social production function theory of Steverink and Lindenberg [3]. Older people want to be independent for as long as possible and be connected to other people, to a neighborhood, or to society [6,7,8,9,10,11]. To engage in different activities and to be involved in the lives of others, both intimate and peripheral contacts, contributes to the well-being of older people [7,8,9,10,11]. In a systematic literature review conducted by ten Bruggencate, Luijkx and Sturm [7], reciprocity—as in doing something for others for a community or a society—was found to be an important concept in the social lives of older people. Reciprocity is the glue that binds older people to others, to a neighborhood, and to society. In doing something for others, older people feel independent and have meaningful lives. When a person gets older, while their social needs and the need to fulfil them do not change much, the resources to fulfil these needs do. With a loss of health, mobility, and network members, it gets more difficult to fulfil social needs [3]. Despite the loss of resources, fulfilling social needs remains important and contributes to the well-being of older people.

In Western countries, the emphasis is on the individuals’ own responsibilities for their health and well-being and on activating people to help each other [12]. In countries like the Netherlands, a shift has already been taking place from a welfare state to a society where participation is promoted in the last couple of years. Older people in local communities in the Netherlands are stimulated to actively be part of the community and society. They are encouraged to remain independent in their own homes for as long as possible, to take care of themselves and others, and to actively participate in society. Local governments stimulate this active participation of its older citizens [13]

### 1.2. Social Interventions for Older People

Many interventions for older people to stimulate social connectedness or independence have been developed and implemented worldwide [14,15,16,17]. Though these interventions are aimed at improving their general well-being and, in that way, the quality of life of older individuals, the interventions differ in terms of their specific focus, their target group, their overall organization, and whether or not technology is involved. Cattan et al. [14], in their systematic literature review about health promotion interventions, made a distinction between group (e.g., activities of educational input or social support), one-to-one (e.g., activities of home visits or telephone contact to provide information, services, or support), service provision (e.g., transport, medical intervention), and community development (e.g., social activities) interventions. The interventions in this study were all designed for older people at risk of being lonely or socially isolated. A few of these interventions used a form of technology.

Though many interventions have been developed for older people, evaluations of these interventions are scarce. Moreover, interventions that have been evaluated often fail to be proven successful [14,15,16,17]. There are several reasons why interventions do not meet the needs of the target group. Many interventions focus on stimulating contacts, and, while this can be beneficial, it does not always match the specific social needs of an older person [16,17,18,19,20]. For instance, in a scoping review of O’Rourke, Collins [19], various interventions to promote social connectedness and prevent loneliness were analyzed. The conclusion of this research was that different approaches other than stimulating social contacts are needed. The authors especially mentioned purposeful activity as an interesting focus. Due to the diversity of the population of older people and the diversity of their social needs, interventions do not always match these needs [7,16]. Therefore, a single intervention that will work for every older individual is simply impossible. Besides the heterogeneity of older people, some older people do not actively participate and are, in a way, withdrawn from society. They are, in fact, invisible and therefore do not actively make use of available interventions. Alternatively, they purposely choose to let social needs go unfulfilled; they choose not to participate [16]. These aspects should also be considered in developing and implementing interventions. From our perspective, interventions should be tailor-made and based on knowledge about social needs and how older people fulfil these. With this study, we developed, implemented, and evaluated an intervention aimed at fulfilling the social needs of older people. Evaluating the intervention can provide deeper insights about the social needs of older people.

### 1.3. Samsam: An Intervention to Fulfil Social Needs of Older People

In 2017, eleven students from the Dutch Design Academy in Eindhoven (DAE) were asked to design and implement an intervention that matched the social needs of older people based on the results and recommendations of previous studies [6,7,8,9]. One of the students developed an intervention that matched well with the results and considerations of these studies. To develop the intervention, this student wanted to gather more information about the population of older people, and she came in contact with an older woman in a residence for older people. For two months, she observed the everyday situations occurring at this residence while having conversations and playing scrabble with the older woman. The student observed that there were many organized activities in the residence, but none of them had a genuine meaning or purpose; the activities did not, for example, focus on the talents or skills of the older people. She also observed that the staff (the caretakers) were working hard and had tight schedules, leading her to the conclusion that it would be important to develop an intervention that was meaningful for older adults with little burden on the staff. The concept of SamSam was born when the student saw an announcement for volunteers to teach the Dutch language.

Samsam is a language café where older people help expats, migrants, and refugees (hereafter referred to as students) in the Netherlands to learn the Dutch language. It takes place once a week, on Tuesday at 14.00 h. The first meeting of Samsam was on the 6th of March 2018. The older people who participate as volunteers are community-dwelling residents and live independently in or nearby the care center. To volunteer in Samsam requires good cognitive and verbal skills. The coordinator of the care center decides which older people meet these requirements. The foreign students (who want to learn the Dutch language) are recruited primarily through Facebook. Their age varies from 20 to 60 years old. They are from countries such as Syria, China, Hungary, Italy, and Turkey.

During the weekly language café, the volunteers and the foreign students engage in conversations about daily life, do short (grammatical) exercises, and play games centered around specific themes. Examples of themes are ‘spring,’ ‘the news,’ and ‘family.’ Every week, supported by informal learning materials, new themes are introduced. For example, for the theme ‘food,’ there were placemats with different topics and questions the volunteer could ask. With this theme, the students and the participants were also asked to bring food from their own culture and country (see Figure 1). The themes activate communication and conversations, and they sometimes relate to the needs of the foreign language learners, with such themes including ‘applying for a job’ or ‘how to build up a network.’ The learning materials help the participants to ask questions and talk about certain topics related to a theme. Samsam takes place in an informal setting—the restaurant and café in a care center in a city in the south of the Netherlands.

The concept of Samsam is not new. There are similar initiatives where volunteers teach foreigners a language in libraries and community centers. Samsam is different in that only older volunteers (aged over 60) are involved and that it is located in a residence for older people and therefore easily accessible for older volunteers. The learning is informal, as there are no formal language or grammar workbooks; only themes are presented, and matching designed materials are used. Samsam is coordinated by a student from the Dutch Design Academy and students studying applied psychology under supervision of professors of the psychology department. Another aspect of Samsam that makes it stand out from similar initiatives is that this intervention has been developed and implemented on the basis of strong empirical evidence regarding the social needs of older people.

### 1.4. Purpose of Research

In this study, we aimed to find out how and to what extent this intervention, in the form of volunteer work and based on the concepts of connectedness, independence, and meaningfulness, supports the participants in the fulfilment of their social needs. Therefore, our research question is:
“What are the experiences of the older volunteers participating in Samsam, and how does participation affect their social needs?”

## 2. Materials and Methods

Following approval by the Tilburg University Ethics Review Board (ERB) EC-2018.EX105, data were collected. We evaluated the intervention in a qualitative way, because we wanted to know the experiences and stories of the older participants regarding the intervention. Qualitative methods can play a significant role in intervention evaluation. Qualitative methods can yield information with a breadth and depth that is not possible to achieve with quantitative approaches [21]. We held two group interviews because of the benefits and possible valuable results of the interaction and discussion between the participants. In addition, we carried out one individual interview, because this volunteer was not able to attend the group interviews.

### 2.1. Participants

The inclusion criteria for the participants were:-Volunteers of Samsam (in total, 10 participants are involved in Samsam)-Participated in three or more of Samsam sessions

In addition, all the participants had good verbal and cognitive skills, because these are requirements for taking part in Samsam.

From the 10 older volunteers, seven agreed to participate in the evaluation. Two participants did not want to participate because they were too busy with family and other activities, and they did not feel like it. One of the female participants could not participate due to her poor health at the time of the data collection.

The group interviews were facilitated by a researcher who has a background in psychology. The facilitator guided the process and ensured that every participant had a chance to speak. The individual interview was conducted by the same researcher.

### 2.2. Procedure

The study consisted of two group interview meetings and one individual interview. Each group interview was with three participants of Samsam.

In June 2018, we telephoned the older people and asked whether they were interested in participating in the evaluation of Samsam. When the participants agreed to take part, they received a letter with detailed information about the study and the group interview. We informed them about the voluntariness of their participation and their right to cancel at any time without giving a reason. They were also asked to complete and sign a letter of consent.

### 2.3. Materials

Our topic list was quite open (semi-structured) and was about the participants’ experience of Samsam. The topic list was partly based on our previous studies about the social needs of older people [6,7,8,9]. We asked questions in relation to bonding with the foreign students, for what reasons they participated, and if and why they would recommend Samsam to other older people. We specifically asked about special moments where they enjoyed or learned something, and how they experienced the atmosphere in Samsam.

### 2.4. Analysis

With the permission of the participants, the interviews were digitally audio recorded and transcribed verbatim. A thematic analysis [22] was employed. Using a qualitative data analysis software (Atlas.ti version 8), inductive codes were attached to quotations relevant to the research question. Two researchers were involved in the coding process to ensure inter-rater reliability [23]. Each transcript was independently coded by two researchers who, to reach a consensus, discussed their coding. Afterwards, the two researchers discussed the codes and relevant themes. We especially looked at social needs and in which ways the intervention helps to fulfil these, but we also looked at the overall experiences and stories regarding the intervention code.

## 3. Results

### 3.1. Sample Description

In total, seven people participated in the interviews, three of whom were female (see Table 1). Some common characteristics of the seven participants were that they have quite socially active lives and that some of them also engage in other forms of volunteer work. They all have relatively good verbal and cognitive skills, which are required conditions to work as a volunteer in Samsam.

In our interviews, the general evaluation of Samsam was a repeated topic in the discussions. Participants mentioned what they like and do not like about the organization and conditions of Samsam. Our first theme was: (1) The general experience of Samsam. On a deeper level, we looked at which social needs are fulfilled in Samsam, and we found: (2) Connectedness and (3) meaningfulness and status to be relevant themes in the transcripts. Each of the three themes is discussed in more detail in the following sections.

### 3.2. General Experience of Samsam

During the interviews, participants often expressed their experiences with the language café in terms of what they like and dislike about this intervention. The participants all like the fact that Samsam has a low threshold for them to participate and is easily accessible. At the time of the interviews, none of the participants had quit Samsam, and they had all volunteered for a couple of months—most participants about 10 times (Samsam takes place every week). Some could not always come because of health reasons or vacation, but, among the participants, there was no intention to stop this work. The participants liked the fact that participating in Samsam is fun and not obligatory. The participants appreciate the themes around which the conversations take place. They found it important to have some guidelines and structure in the form of a theme (e.g., there are themes related to spring, food, or travelling). The participants have liked the different themes, except for one time when the theme was ‘design and art.’ The participants discussed pieces of art and design with the students. Most of the participants did not enjoy this. One participant explicitly said:
“When it is going to be like this, I quit.”*(Woman, 80)*

Apparently, this theme did not match their interests. The participants also found it important for the students to be motivated. One time, one participant had a student who was not motivated and mentioned this student during the group interview:
“When you don’t want to learn anything, you better stay away.”*(Man, 92)*

Some of the participants found participating in Samsam to be hard work, and, although they enjoyed doing it, it sometimes cost them too much energy. One participant felt quite exhausted after participating in Samsam. One other participant said:
“I like doing it, but it is hard work.”*(Man, 84)*

The participants indicated that they would recommend participating in Samsam to other older people. One of the female participants was introduced to Samsam by a friend. That Samsam has a low threshold, is nearby, is easy accessible, and is a lot of fun were reasons for her to join. One of the participants said:
“From all the volunteer work I do and have done, I like this (Samsam) the best.”*(Man, 88)*

The fun element is essential in Samsam. The participants indicated that they enjoy the laughter and cozy atmosphere.

### 3.3. Connectedness

The need for connectedness to other people and to society was a major theme during the interviews. The participants of Samsam find friendship and companionship with their students and with each other. Some of the participants meet with each other outside Samsam. One of the female participants enjoyed the company of another male participant and joked about his obsession with planes and flying. They know each other from living in the same neighborhood, and their contact is now stronger through Samsam. She stated that the two of them regularly go for little walks together, and they talk about all kind of things, including Samsam.


*“We know each other, we walk together, and we make jokes together.”*
*(Woman, 80)*

The participants bond with the students through their shared interests. One of the participants is a retired architect. In Samsam, he teaches the Dutch language to a Hungarian female architect. They are friends now and made a connection through their shared interests and background. They also meet outside the regular Samsam times to drink tea. The participants often mentioned the laughter and fun they have. One of the first remarks one of the participants made was:
“What more can you ask for, at age 88, to have that much fun.”*(Man, 88)*

The students learn from the participants but also vice versa. The participants learn about other cultures, habits, and countries. The sharing of stories is key to the success of Samsam. The connectedness between the participants and students is influenced by the feeling of reciprocity the participants have with their students. The participants and students talk about common interests, hobbies, and experiences. One of the participants said:
“They also ask me questions about my life and my interests.”*(Woman, 61)*

The contact in Samsam is not a one-way direction of learning and communication. The participants indicated that there is a sincere interest in one another. This means that the relationship is reciprocal. The good atmosphere contributes to the connectedness the participants feel. The participants describe the atmosphere in Samsam as fun, cozy, and open.

The participants all show openness to other people and cultures, and they have a will to help others. The older participants learn about other cultures and countries, and they are often fascinated about the differences in culture. One participant has talked with his Chinese student about how small the Netherlands is in comparison with China. They talk about differences in habits, norms, and values. Samsam seems to create an intercultural understanding between the two groups—the older participants and the students. The foreign students are, in most cases, much younger than the participants, so besides an intercultural understanding, an intergenerational understanding and bonding is also present. The older generation bonds with the younger generation, thus creating understanding and respect. As one of the participants said about Samsam:
“You connect to each other human to human.”*(Woman, 83)*

The participants and students have an open attitude towards each other and, in this way, connect to each other. They also learn from each other. One of the students is a man from Syria who impressed one of the participants with his optimistic and positive nature. The participant said:
“I learned from him that they are not sad or pathetic despite the situation they come from.”*(Woman, 61)*

The connectedness is not only felt towards the students. By participating in Samsam, the participants feel connected to society. In the interviews, they demonstrated a connectedness and a will to contribute to society. They want to contribute to a better world because they feel connected to this world. This relates strongly to the feeling of meaningfulness and status, which will be described in more detail in the next paragraph.

### 3.4. Meaningfulness and Status

Many of the participants addressed the need to be meaningful, to be respected, and to have status. The participants are happy and proud that they can teach something to the students. They all have a need to stay active and contribute in a meaningful way; this was explicitly or implicitly mentioned by almost all of the participants. One said:
“I want to do my bit in the world.”*(Man, 92)*

This participant, an active and eloquent man of 92 years old, mentioned in the group interview that he sees people of his own age and younger being passive and looking tired. He mentioned that he sometimes wants to tell them to get up and do something useful, because that is what keeps you young. He also argues that some people need a little push to get involved in initiatives like Samsam. He made remarks about using your talents and staying active even when you are old:
“You have to stay close to who you are and see what (activity) matches with that.”*(Man, 92)*

To help others and teach something to other people was found to be the main reason for participating in Samsam. Furthermore, the motivation to participate varies from fun, to political, to social reasons, and to a combination of reasons. Some of the participants have a strong social motivation to participate. One of the participants, a very involved and friendly man of 84, has deep sympathy for refugees and migrants. He sees a big problem in the fact that foreigners cannot find a job because they cannot speak the Dutch language. The participants all sympathize with these refugees and foreigners, and, in teaching them the Dutch language, they hope that there will be more and better (job) opportunities for them.

Being part of Samsam gives the participants meaning, purpose, and status. Status refers to using your skills and assets, being taken seriously, and being respected. The participants are proud that these students come to them every week and that they see progress with their students. One of the participants said:
“I had never thought that at my age I would teach someone the Dutch language, and that I would be successful in it!”*(Woman, 80)*

The themes, materials, and assignments are present every week, but the participants all have their own style and manner of teaching. In this way, they use their own talents and their skills to teach. One of the participants lets the students write everything down. Another participant gives homework: His/her students must learn five new words a week. The participants also really want to do a good job and want to make progress with the student sitting across the table. One participant said:
“It is quite of exciting too, because you want to do a good job.”*(Woman, 83)*

The participants are proud and happy that they can use their talents in Samsam. One of the participants always had a passion for language. Samsam is an opportunity for her to use her talents and passion. She said:
“Languages are my passion and I thought that (Samsam) was really something for me.”*(Woman, 80)*

The need to be meaningful is linked to the need of status and independence. Being taken seriously and treated with respect by using your assets, talents, and skills makes a person feel autonomous and independent. In doing something for another person or contributing in one way or another, a person can feel less dependent and more equal. Most of the participants have had busy lives, careers, and a prominent role in society. Many of the participants find it important that they can use their talents and skills. For instance, one of the participants has worked in different disciplines of volunteer work. He proudly mentioned that he has become an honored citizen of the city where he lives. He is chairman of a country club and has a lot of different functions. He is sharply dressed and highly educated. In his working life, he was respected, and in his retired life, he achieves the same by staying active and doing volunteer work like Samsam.

The participants enjoy speaking with the students about their lives and passions. One of the participants has lived in various exotic countries, and he very much enjoyed exchanging experiences about these countries and their politics. One of the participants said:
“Then we (the student and I) talked about Slovakia and how corrupt it is out there.”*(Man, 92)*

As discussed in the previous section, the participants all want to contribute in a significant way and want to mean something. When asked specifically about their main motivation for participating in Samsam, one of the participants said:
“About my motivation, my reasons, I want to mean something to other people.”*(Man, 92)*

The participants all have a strong motive to help other human beings. One of the participants is engaged in a lot of different forms of volunteer work. When asked why she is so active and committed, she said:
“*I have a need to bring something to other people and to take care of them.”**(Woman, 80)*

Besides Samsam, she also works with older people with dementia. She loves the fact that despite their cognitive problems, they are happy to see her. She said:
“The nurses are happy, I am happy, and the patients are happy. What more do you want.”*(Woman, 80)*

In doing something for other people, they also do something for themselves. By using their talents and skills to contribute in a meaningful way, they feel proud and satisfied—therefore, important social needs such as status and meaningfulness can be fulfilled.

## 4. Discussion

This article describes the development, implementation, and evaluation of an intervention that fulfils the social needs of older people: Samsam. The evaluation furthermore gives deeper insight into the social needs of older people and the way they can be fulfilled. Samsam is a language café where older people teach foreign students the Dutch language. This intervention is based on our previous insights related to the important social needs of connectedness, meaningfulness, and independence. With this study, we wanted to find out how participants experience this intervention, and in what way participation in Samsam contributes to fulfilling their social needs.

We found that participating in Samsam has generated a lot of positive reactions from the participants. It has positive side effects because it creates intergenerational openness and intercultural understanding. It also has a fun element to it, has practical benefits, it is easily accessible, and is organized by students. Volunteers find it worthwhile to participate in Samsam despite the hard work, skepticism about some of the weekly themes, and non-motivated students. Results indicate that Samsam offers opportunities for the participants to be connected in different ways, as they are connected to other people and to society. The main reason the older people volunteer in Samsam is that they want to mean something to other people and to themselves. On the basis of the overall positive experiences of the participants, we can conclude that this intervention seems to match the social needs and, in that way, contributes to the well-being of the participants. This matches the findings of O’Rourke, Collins [19], where engaging in purposeful activities was seen as a promising factor in interventions to stimulate social connectedness. It is also in line with our previous research, where we found that interventions that support older people in giving back and doing something for other people, for a community, or for society in general can be and are successful [6]. A person can feel useful, independent, and connected by doing volunteer work [6,11,19,24]. Engaging in volunteer work can therefore fulfil different social needs for older people and, in that way, contribute to their overall well-being [24,25]. Volunteer work is therefore seen as a promising activity and a possible successful intervention for older people.

Linked to the social production function theory of Steverink and Lindenberg [3], this study illustrates that Samsam supports older people in fulfilling all three social needs: Affection, behavioral confirmation, and status. Status is perhaps the most interesting need, because this is rarely linked to older people. According to Steverink and Lindenberg [3], status is fulfilled by relationships that give you the feeling that you are being treated with respect, are being taken seriously, and are known for your achievements, skills, or assets. It has a strong relationship with feeling independent and autonomous. Most interventions focus on stimulating contacts between older people and therefore focus primarily on fulfilling the need of affection. Older people want to be actively involved in what happens in a society or community [6,26]. They do not want to stand on the side lines and do nothing. They want to actively contribute to a better world. Samsam enables older people to actively contribute something to the lives of refugees and migrants. In this way, the intervention contributes to the well-being of both groups.

The participants in our study clearly do not disengage from society. They have a strong need to be involved and engaged. This seems contradictory to the classical disengagement theory of Cumming and Henry [27], who proposed that in the normal course of aging, people gradually withdraw or disengage from social roles. As also stated by [28], this theory made sense in the 1960s, but it seems outdated today. In current society and modern times, a new way of looking at older people is needed. The social selectivity theory (SST) of Carstensen, Fung, and Charles [29] states that the social networks of older people are formed through network movements that are characterized by a process of selectivity, and that these movements are motivated by the emotional goals of the older individuals. The SST states that older adults maintain or increase their interactions with family and intimate friends, and they invest less in peripheral network members. The SST also argues that old age is associated with an increasing motivation to derive emotional meaning from life and a decreasing motivation to expand one’s horizons [29]. This partly corresponds with the results of our study, as the participants do indeed search for purpose and meaning, but they also expand their horizon by participating in Samsam. In Samsam, they also connect with more peripheral members—the students of Samsam. As also indicated in the systematic review of Bruggencate [7], both intimate and peripheral relationships are and remain important for older people.

In general, most (local) communities in Western societies have policy makers that work on improving the social wellbeing of older people by developing and implementing interventions. Developing and implementing interventions is a way to work on improving the social wellbeing of older people. The way a society looks at older people has an influence on how older people are treated within this society. Diamond [30] reported, for example, in his book “The World Until Yesterday: What Can We Learn from Traditional Societies?” how different societies deal with their older citizens. He pointed out that in Western society people are respected when they have work, are independent, and are self-reliant. In addition, youth and being young is being idealized. In some more primitive societies, older people still have important (social) roles; they make baskets, cook, and look after grandchildren. Diamond’s advice is therefore that older people should play a more active role in Western societies. The life experience and wisdom of older people can be more efficiently used and put into practice. [30].

There are different perspectives about the role of older people in society and whether they have to keep busy, stay active, and contribute to society. Some argue that an older person should not have to perform this busy role [31,32,33]. Holstein and Minkler [32] and Martinson and Minkler [33] argued that there should not be an economic or political gain in keeping older people active, and older people must not be misused or be seen as a resource. Furthermore, by only pointing out the positive aspects of (active) ageing, older people who cannot live up to this standard are being mistreated and overlooked [32,33]. This study and previous studies [6,7,8,9] indicate that, for a group of older individuals, it is important to be have meaningful lives—contributing to others, to a neighborhood, or to society can fulfil this need [8]. Every older person must be free to live the life they want and need, so interventions should be tailor-made and meet the specific social needs of the older individual.

As is indicated in the study of Steverink and Lindenberg [3], the social needs of people stay the same during their lives, but the resources to fulfil them change. With a loss of health, mobility, and network members, the fulfilment of social needs becomes more difficult when growing older. This intervention, which is primarily based on verbal, communication, and some didactic skills, is in most cases suitable for older people. It is not a physically difficult task, and it takes place in the care center. It is an easily accessible and achievable intervention for older people in different care centers. Besides the fact that a large group of older people can potentially be part of Samsam or comparable initiatives, we realize that being a volunteer in the language café is not suitable for every older person. There is simply not one intervention that matches with the social needs of every older person [7,16]. Perhaps the most vulnerable older people do not benefit from interventions like this. The participants have to be in a cognitively and verbally good state. It also appears that our participants have an open mind and a will to help others. This intervention can therefore appeal more to older people who have an open mind and a will and motive to help other people.

The group of volunteers currently involved in Samsam consists of ten older people, from which seven agreed to share their experiences with us in two group interviews and an individual interview. We realize this number is small, and we must be cautious with generalizing our findings. However, our findings underline a lot of existing evidence which point out the benefits of volunteer work [6,24,25,34]. Our study also reveals some new insights, such as the importance and possibility for older people to fulfil their need for meaningfulness and status. We also found that openness is a promising personality trait which can be beneficial for the well-being of older people because it can enhance social connectedness.

## 5. Conclusions

Based on our results, we may conclude that Samsam is an intervention that helps older people fulfil their social needs. As the participants contribute in a meaningful way, they probably connect more easily. The connection towards others and to society is also a result of their openness. When you connect, you are not isolated, and that connectedness contributes to the overall well-being of people. Connectedness is better achieved in a reciprocal relationship, such as in a relationship where the older individual contributes in a meaningful way. We believe that interventions that focus more on purposeful activities or volunteer work also create connectedness and have more positive effects on the general well-being of older people. Every older person has talents, skills, and life experiences which can be put to use. Interventions should appeal to the needs of older people to meet, to matter, and to have fun, as well as their needs for connectedness, meaningfulness, and independence.

## Figures and Tables

**Figure 1 ijerph-16-02307-f001:**
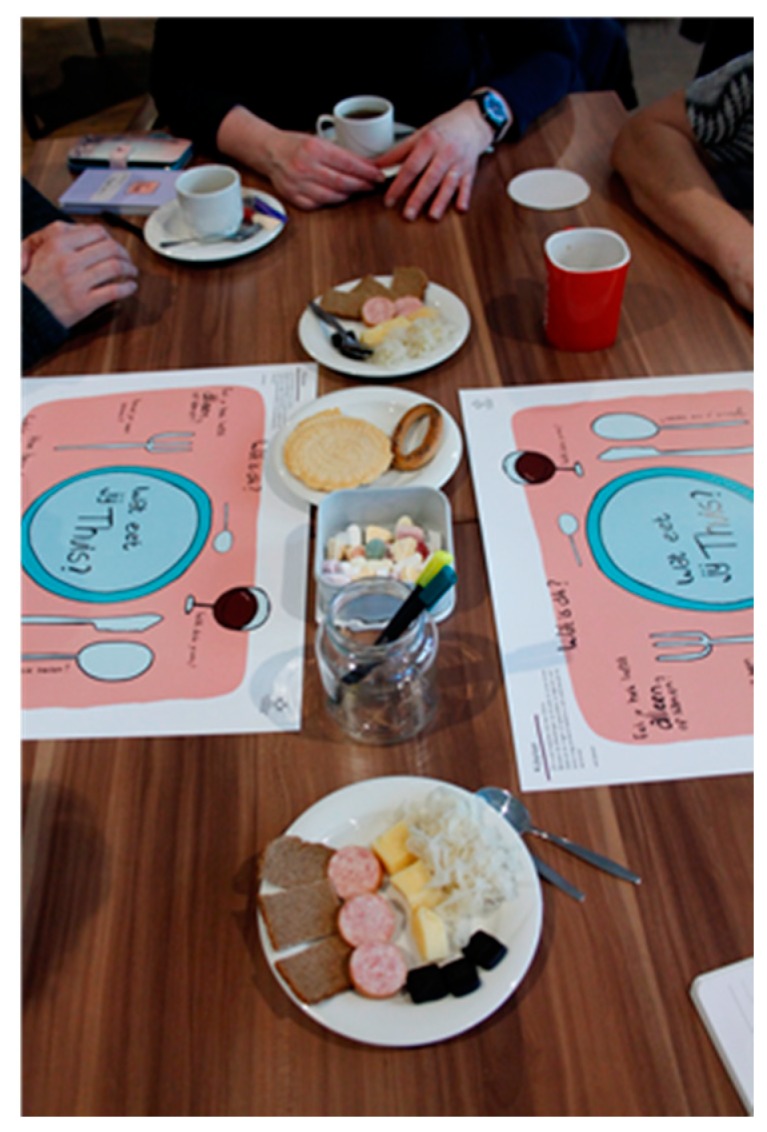
Example of materials used in Samsam.

**Table 1 ijerph-16-02307-t001:** Age and sex of the participants.

Group Interview 1	Group Interview 2	Individual Interview
Woman, 61	Woman, 83	Woman, 80
Man, 88	Man, 92	
Man, 84	Man, 84

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
