# Peer review of "To Meet, to Matter, and to Have Fun: The Development, Implementation, and Evaluation of an Intervention to Fulfil the Social Needs of Older People"

_ijerph, 2019, doi:10.3390/ijerph16132307_

Round 1

Reviewer 1 Report

This is an interesting study and the authors present material that is important and with great implications to practice. There are a couple areas which can be improved to enhance the overall quality of the paper. 

First, the introduction needs enhancement, to attempt to contextualise the knowledge offered. Provide some information about the current socio-political support to older people in this context, to ensure that the reader can position the results accordingly. 

Also, the analysis can be enhanced with more links with the findings, as well as theoretical frames. I wonder if the authors are aware of the disengagement theory? This would truly support the analysis of this paper. 

Author Response

Thank you very much for your comments and your valuable contribution. In the attachment you will find your comments and the adjustments we made in combination with those of the other reviewer. The revised manuscript is submitted.

Reviewer 2 Report

Comments to the authors

A brief summary

The aim of this paper “To Meet, to Matter and to Have Fun: the Development, Implementation and evaluation of an Intervention to Fulfil the Social Needs of Older people” is to describe development, implementation and evaluation of an intervention in which older volunteer workers taught Dutch to the expats, refugees and immigrants. The intervention was based on theoretical background and the concepts of connectedness, independence and meaningfulness. It aimed at fulfilling social needs of older people. An essential part of the paper is to describe results of a qualitative study on how and to what extent the intervention was able to fulfil these social needs of older participants. Evaluating the intervention provides deeper insights about the social needs of older people, and thus the study and the paper contribute to the literature by adding understanding on significance of especially meaningfulness and status to older people.

Broad comments

The paper include all relevant issues to enable reader to understand what was done and how the results and the conclusion were achieved.

Theoretical background is good and relevant. By raising Rowe and Kahn’s successful aging model, Steverink and Lindenberg’s social production function theory, Bruggencate’s earlier study of social need relevant for older people, and Bruggencate & Luijkx’s systematic review highlighting reciprocity, authors present theoretical background for the paper and the intervention.

Qualitative approach is used as a means to evaluate the intervention. Qualitative study is important in enabling experiences and stories of older volunteers, and thus receiving deeper understanding of social needs of older people and the ways the intervention is able to fulfil them.

Themes seems relevant and justified based on the interview data. Understanding/keeping up the flow of presentation of the results is slightly hard.

In the discussion, the authors deal with the results of the study in general, and specifically they highlight the most important aspects. Older people, still when aged, need to feel needed, useful, and respected. Reciprocity is important as well. The SumSum intervention is an example how the social need of status can be fulfilled. However, as authors indicate, one intervention cannot be suitable for all older people, and thus tailoring and various interventions are needed. Restrictions of generalizing the results were discussed.

Author Response

(The authors gave the same response as above.)
